# ID-to-3D: Expressive ID-guided 3D Heads via Score Distillation Sampling

**Francesca Babiloni, Alexandros Lattas, Jiankang Deng, Stefanos Zafeiriou**
Imperial College London, UK
https://idto3d.github.io

## Abstract

We propose ID-to-3D, a method to generate identity- and text-guided 3D human heads with disentangled expressions, starting from even a single casually captured in-the-wild image of a subject. The foundation of our approach is anchored in compositionality, alongside the use of task-specific 2D diffusion models as priors for optimization. First, we extend a foundational model with a lightweight expression-aware and ID-aware architecture, and create 2D priors for geometry and texture generation, via fine-tuning only 0.2% of its available training parameters. Then, we jointly leverage a neural parametric representation for the expressions of each subject and a multi-stage generation of highly detailed geometry and albedo texture. This combination of strong face identity embeddings and our neural representation enables accurate reconstruction of not only facial features but also accessories and hair and can be meshed to provide render-ready assets for gaming and telepresence. Our results achieve an unprecedented level of identity-consistent and high-quality texture and geometry generation, generalizing to a "world" of unseen 3D identities, without relying on large 3D captured datasets of human assets.

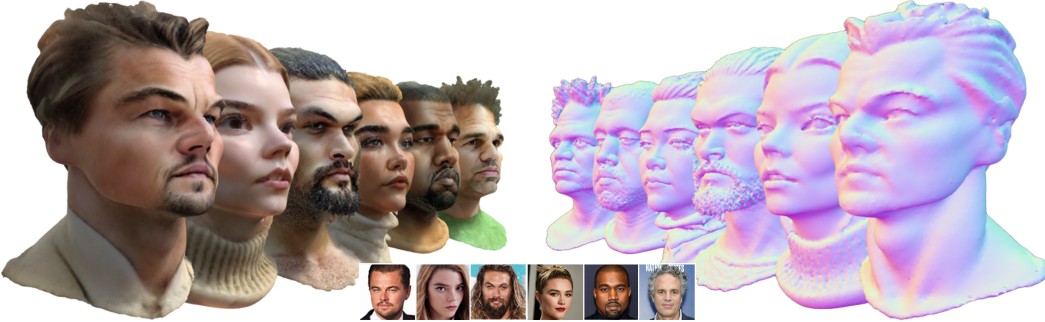

Figure 1: ID-to-3D leverages identity conditioning and score distillation sampling on large diffusion models, achieving high-quality 3D human asset generation from "in-the-wild" images, without training on large scanned datasets. From left to right: a) renderings, b) input images, c) normals.

## 1 Introduction

The remarkable ability of humans to discern facial characteristics and emotional cues in others makes the development of high-quality 3D head avatars a challenging yet foundational task for a diverse array of emerging applications, including digital telepresence, game character generation, and the creation of virtual and augmented reality experiences. However, the acquisition of such 3D human assets remains a daunting task, that requires either manual work typically performed by graphic artists, or expensive and laborious scanning. High-quality 3D facial scanning, achieved initially

38th Conference on Neural Information Processing Systems (NeurIPS 2024).

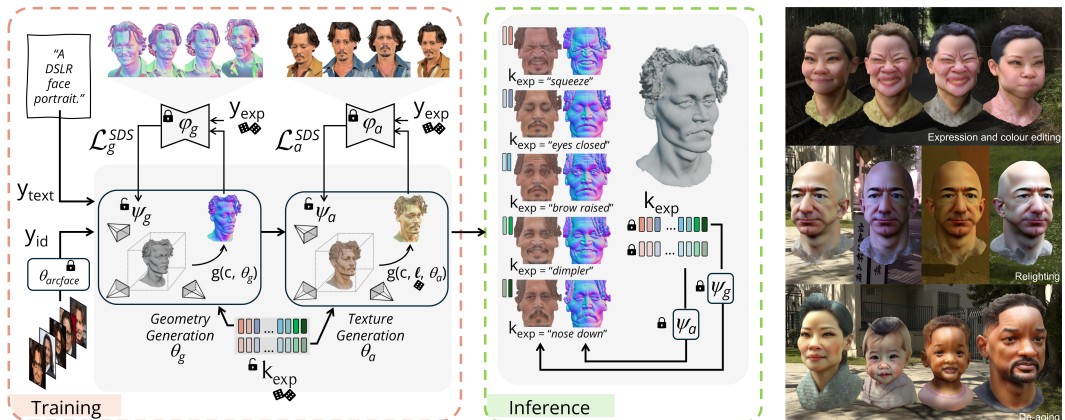

Figure 2: **(Left) Overall pipeline.** ID-to-3D generates expressive 3D head avatars via ArcFace $y_{id}$ and textual $y_{text}$ conditioning. It uses as prior geometry-oriented $\phi_g$ and albedo oriented $\phi_a$ pretrained models. **Training)** The training phase uses SDS to optimize 3D geometry $\psi_g$, texture $\psi_a$, and a set of expressions latent codes $k_{exp}$. It also leverages random lighting $l$ and random expression conditioning $y_{exp}$. **Inference)** At deployment time, ID-to-3D extracts high-quality identity-aware expressive 3D meshes. **(Right) ID-consistent expressive 3D heads** generated by our method. ID-to-3D creates 3D assets that support relighting, ID-consistent editing, and physical simulation.

by hardware with controlled polarized illumination [15, 51, 23], has been simplified tremendously using simpler devices, color-space methods, and inverse rendering [25, 39, 3]. Nevertheless, the requirements for expert knowledge, hardware, and computational time do not allow wide adoption or mass usage.

Consequently, statistical modeling and the rise of deep learning investigated techniques to reconstruct 3D face assets from casually captured images of a subject in arbitrary poses, lighting conditions, and occlusions (referred to as "in-the-wild"). 3D Morphable Models (3DMMs) and Generative Adversarial Networks (GANs) can be used to model facial geometry [4, 19, 56, 24], while GANs and diffusion models have achieved state-of-the-art modeling and reconstruction of facial appearance [40, 22, 92, 60].

However, a common denominator in all the above is the requirement for vast datasets of scanned facial shapes and appearances, in order to avoid limited generalization, ethnicity under-representation, and oversmoothed geometries, requiring up to $10,000$ scans for 3DMMs training [6] or facial appearance modeling [22]. Moreover, the requirement for aligned data in 3D, and also in UV texture maps, limits the utilized area to the facial region, and introduces registration errors and additional costs. Our proposed method bypasses these significant issues by utilizing large 2D generative models, pre-trained on vast and easy-to-acquire 2D images, and using only a small set of rasterized 3D data for finetuning, while achieving state-of-the-art 3D head generation.

More recently, the use of Score Distillation Sampling (SDS) [63] and large-scale diffusion models explored the automatic generation of 3D content thanks to text [85, 10, 45, 44, 31, 92, 37, 26] or image prompts [66, 91, 64, 79, 80]. Despite promising results, leveraging 2D priors to generate realistic 3D head avatars remains challenging: **(1)** Large text-to-image foundational models are usually trained to generate realistic RGB images, where geometry, texture, and lighting cannot be individually segregated. This narrow "rendering knowledge" in the 2D guidance compromises the 3D fidelity, texture quality, and consistency of the generated assets, leading to unrealistic geometries and distorted textures with recognizable artifacts such as Janus problems, incorrect proportions, oversaturated albedo, and mismatching texture and geometry details. **(2)** It is difficult to precisely control facial attributes solely using typical prompting methods [13, 59, 88, 84]. Textual prompts lack the granularity to single out the specificity of a subject's identity and facial expression that might be lengthy and complex to convey in natural language. Moreover, methods that leverage image prompts lack the capability to capture features that represent the facial identities of a subject independently of pose, expression, or contextual scene information (i.e., ID embeddings). The lack of reliable control over identity and facial expression prevents the personalization of 3D head avatars, resulting in a limited range of possible output expressions or custom attributes without incurring identity drifting.

In this work, we present ID-to-3D, a new method to generate expressive and identity-consistent 3D human heads using a text prompt and a small set of 1-5 casually captured, in-the-wild, images of a subject. ID-to-3D creates a variety of separated yet ID-consistent expressions in a single optimization, leveraging as priors compositionality and task-specific 2D diffusion models. To ensure consistency, the identity of each subject is encoded via facial embedding features and, to encourage expressivity, emotions are disentangled via a novel ID-specific neural parametric representation. The generation of each 3D asset leverages score distillation sampling and a two-stage pipeline to create shape details, texture features, and materials. During optimization, the guidance is given by a foundational model extended into an a) identity-aware, b) task-aware and c) expression-aware variant, working as 2D prior for either geometry or texture generation. Each stage guidance is trained only once for all the potential identities, with a lightweight fine-tuning strategy changing only $0.2\%$ of all the available training parameters.

Overall, we present the following contributions: **(1)** *ArcFace-Conditioned 3D Head Asset Generation*: we introduce the first method for arcface-conditioned generation of 3D head assets using SDS. **(2)** *Novel ID-Conditioned Expressive Model*: our model creates an identity-conditioned expressive representation for each subject, enabling the generation of up to 13 unique and ID-consistent expressions captured by latent codes and associated with a set of 3D assets with separate geometric, albedo, and material information. **(3)** *Novel Text-to-2D-Normals and Text-to-2D-Albedo Models:* we present a novel approach to creating ID-conditioned and expression-conditioned text-to-image models capable of generating realistically plausible normals and albedo images from a small set of 3D assets. Extensive experiments confirm that our method outperforms text-based and image-based SDS baselines by producing relight-able 3D head assets with unprecedented geometric details, superior texture quality, and able to exhibit a wide variety of expressions. Fig. 1 displays 3D heads generated with ID-to-3D.

## 2   Related Work

**3D Human Generation and Reconstruction.** Human modelling and reconstruction in the 3D domain is typically based on 3D Morphable Models (3DMMs) [4, 6, 43, 78], which model variations in human shape and appearance using PCA. The advent of deep learning enabled very expressive reconstructions [14, 68], but also extended this line of research beyond linear spaces. Neural Parametric Head models [24] explored the use of signed distance function and deformation field to generate expressive geometries. Generative Adversarial Networks have also been proposed to generate [42, 21] and reconstruct faces from "in-the-wild" images [22, 50, 40]. Diffusion models showed impressive results in modeling skin textures [92, 60] especially when paired with large-scale, high-quality datasets of real scans.

**Text-to-3D Human Generation.** The creation of 3D human assets via text conditioning has seen significant progress, building on the foundations laid by advances in text-to-2D generation [65, 69, 72, 67]. Initial attempts [86, 29, 54, 73, 55, 32, 11] utilized the CLIP language model to optimize implicit or explicit 3D representations. The seminal work of DreamFusion [63] introduced the Score Distillation Sampling (SDS) loss, which uses a pre-trained 2D diffusion as prior for 3D generation. This work led to a revolution in text-to-3D generation [53, 57, 27, 85, 45, 48, 52, 47, 12] and text-to-3D human generation [9, 76, 45, 53, 26, 46, 49]. DreamAvatar [8], TADA [44] and Headevolver [82] build upon the use of a template [61, 43, 5] to create 3D human avatars with controllable shapes and poses. Fantasia3D [10] separated geometry and texture training, employing DMTET [77] and PBR texture [58] for the 3D representation. HumanNorm [31] introduced the idea of training a normal diffusion model to guide high-quality geometry generation. Despite promising results, these models face challenges in generating highly detailed and expressive 3D heads, due to the inherent limitations of natural language conditioning.

**Personalized Generation with Diffusion Models.** Controllable generation is crucial to develop widely applicable generative models. Work in this domain includes the costumization of GANs [74, 33, 35, 36, 34] and research dedicated to steer the generation of large-scale foundation models with additional control signals [20, 70, 30, 93, 88, 71]. The use of ID embeddings as an alternative to text prompts showed promising results in 2D generation [62, 87, 84, 59], but remains underexplored in the creation of 3D human avatars with SDS. Related research supplements traditional text prompts in SDS pipelines with image-based prompts. DreamBooth3D [66] and Avatarbooth [91] proposed to train an ad hoc model to use as a guide in the generation of 3D objects or avatars. Magic123 [64] incorporate 3D and 2D priors in SDS generation. DreamCraft3D [79] proposed a hierarchical 3D

content pipeline to generate textured 3D meshes from a single unposed image. Despite encouraging results, the creation of high-quality head assets with these methods remains challenging, due to the difficulty of extracting appropriate facial features using only naive image prompting.

## 3    ID-to-3D

We propose a novel method for identity-driven human head generation, which utilizes a pre-trained 2D model to distill expressive head avatars with high geometric details and high-fidelity textures, avoiding the need for large-scale training on 3D datasets. As illustrated in Figure 2, starting from a subject's identity embeddings, our method trains a set of latent expression representations using a two-stage SDS pipeline. After convergence, the learned 3D representation can be used to create ID-driven expressive heads, that are ready to be used in common rendering engines.

### 3.1    3D Head Optimization Objective

A Score Distillation Sampling generation pipeline optimizes a 3D representation $\theta$ using a pre-trained 2D diffusion model $\phi$ as guidance. The pipeline optimization objective is to align the distribution of 3D asset renderings with the target distribution $p(\mathbf{x}_0|y_{\text{text}})$, created by the 2D diffusion model conditioned on an input text $y_{\text{text}}$. Given the distribution of renderings under various camera conditions $q^\theta(\mathbf{x}_0|y_{\text{text}}) = \int q^\theta(\mathbf{x}_0|y_{\text{text}}, \mathbf{c})p(\mathbf{c})d\mathbf{c}$, the optimization objective reads:

$$\min_\theta D_{KL}(q^\theta(\mathbf{x}_0|y_{\text{text}}) \parallel p(\mathbf{x}_0|y_{\text{text}})). \tag{1}$$

The target distribution $p(\mathbf{x}_0|y_{\text{text}})$ is typically estimated by a foundational text-to-image diffusion model that approximates the distribution of natural RGB images [69] (i.e., trained over large and uncurated datasets such as LAION-5B [75]). Despite its indisputable success in creating a variety of assets, using the above objective and guidance to generate detailed 3D heads remains a complicated task. First, this target distribution might drift significantly from the distribution of natural heads rendered in realistic light and camera conditions. Second, a general one-shot guidance model does not have explicit ways to differentiate texture and geometric characteristics, which makes the creation of light-independent texture and high-quality geometry extremely challenging. Third, the naive use of text prompts limits the control over the generated head assets, since textual prompts cannot easily or exhaustively capture facial and expression features. In our pipeline's generation, we decompose the Obj. 1 into two smaller and more controllable objectives:

$$\min_{\theta_g, \theta_a} \underbrace{D_{KL}(q^{\theta_g}(\mathbf{z}_0^n|\mathbf{c}, y_{\text{text}}, y_{\text{exp}}, y_{\text{id}}) \parallel p(\mathbf{z}_0^n|\mathbf{c}, y_{\text{text}}, y_{\text{exp}}, y_{\text{id}}))}_{geometry\ generation\ objective}$$
$$+ \underbrace{D_{KL}(q^{\theta_a}(\mathbf{z}_0^a|\mathbf{c}, \mathbf{l}, y_{\text{text}}, y_{\text{exp}}, y_{\text{id}}) \parallel p(\mathbf{z}_0^a|\mathbf{c}, y_{\text{text}}, y_{\text{exp}}, y_{\text{id}}))}_{texture\ generation\ objective}. \tag{2}$$

In the above equation, $\theta_g$ represents the parameterization of the 3D geometry, $\mathbf{z}_0^n$ denotes normal maps, $\theta_a$ denotes the parameterization of the 3D textures and $\mathbf{z}_0^a$ albedo textures. Conditioning is introduced in the form of textual prompt $y_{\text{text}}$, identity condition $y_{\text{id}}$, and expression condition $y_{\text{exp}}$. The letter $\mathbf{l}$ denotes the lighting condition of the rendered image. The target distributions for geometry and texture generation refer to the ideal distribution of head-specific normal and texture maps, which are in practice estimated via geometry-oriented and albedo-oriented models guided by face-specific conditioning.

### 3.2    2D Guidance

To initiate the 3D head reconstruction, we start from the development of 2D priors capable of accurately separating texture and geometric details while, at the same time, consistently representing the facial characteristic of a subject under various expressions, conveying different emotional states. The difficulty of this task lies in its nuanced nature, exacerbated by the lack of large-scale 3D human scan datasets, which makes the capture of detailed face features and the generalization to new identities particularly difficult when training from scratch or even when naively fine-tuning from a large-scale model. To solve this challenge, we propose to explicitly model geometry and appearance domains, identity conditioning $y_{\text{id}}$ and expression conditioning $y_{\text{exp}}$, achieving a modular separation

of otherwise entangled-together information. To overcome the need for a large-scale dataset, we leverage a small dataset of human heads with different expressions (NHPM) [24], a pre-trained stable diffusion model (SD) [69], and a selective fine-tuning strategy that affects only a minimal number of parameters, needed to accommodate these new conditionings. In practice, we use rasterized normals as a 2D proxy for geometric information and rasterized albedo as a representation of appearance information. We treat the shift from the natural image distribution towards normal maps and albedo textures as "style-transfer" tasks, aiming to leave the content of the SD features unchanged while modifying their self-similarity information. We use Low-Rank adaptation matrices (LoRA) [30] to adjust Query $\mathbf{Q}$, Key $\mathbf{K}$ and Value $\mathbf{V}$ features of the self-attention to work in the adjusted normal and albedo domains. The normal-adapted self-attention equation becomes the following:

$$\mathbf{Z}_{\mathbf{SA}}^n = \text{Att}(\mathbf{Q}^n, \mathbf{K}^n, \mathbf{V}^n), \mathbf{Q}^n = \mathbf{XW}_Q + \mathbf{XW}_Q^n, \mathbf{K}^n = \mathbf{XW}_K + \mathbf{XW}_K^n, \mathbf{V}^n = \mathbf{XW}_V + \mathbf{XW}_V^n \tag{3}$$

while the albedo-adapted self-attention can be read by changing the superscript $n$ to $a$. As identity representation we select the identity embeddings $\mathbf{y}_{\text{id}}$, from a state-of-the-art face recognition network [17, 16, 94], a compact vector of facial features extracted from "in-the-wild" images of a subject. As expression conditioning, we use $\mathbf{y}_{\text{exp}}$ CLIP embeddings [65] for the textual descriptor of the 23 FACS coded expressions proposed in FaceWareHouse [7]. To ensure control over the generated head and face representation during deployment, we treat the integration of identity and expression information in the baseline architecture as "multimodal conditioning", by including their contribution in the SD cross-attention layers via IP-Adapter [88] strategy:

$$\mathbf{Z}_{\mathbf{CA}}^n = \text{Att}(\mathbf{Q}, \mathbf{K}^{\mathbf{text}}, \mathbf{V}^{\mathbf{text}}) + \lambda_{id} \cdot \text{Att}(\mathbf{Q}, \mathbf{K}^{\mathbf{id}}, \mathbf{V}^{\mathbf{id}}) + \lambda_{exp} \cdot \text{Att}(\mathbf{Q}, \mathbf{K}^{\mathbf{exp}}, \mathbf{V}^{\mathbf{exp}}) \tag{4}$$

In the above equation, $\lambda_{id}$ and $\lambda_{exp}$ control the contribution of identity and expression conditioning. The $\mathbf{Q}{=}\mathbf{XW}_Q$ term represents the Query extracted from SD features, $\mathbf{K}^{\mathbf{exp}}{=}\mathbf{y}_{\text{exp}}\mathbf{W}_K^{\text{exp}}$ and $\mathbf{V}^{\mathbf{exp}}{=}\mathbf{y}_{\text{exp}}\mathbf{W}_V^{\text{exp}}$ the Key and Values extracted from the expression embedding, and $\mathbf{K}^{\mathbf{id}}{=}\mathbf{y}_{\text{id}}\mathbf{W}_K^{\text{id}}$ and $\mathbf{V}^{\mathbf{id}}{=}\mathbf{y}_{\text{id}}\mathbf{W}_V^{\text{id}}$ the Key and Values extracted from the identity embedding. We train only our additional parameters, leaving the rest of the model frozen, targeting $0.2\%$ of the overall trainable parameters. We separately optimize a 2D prior for geometry $\phi_g$ and textures $\phi_a$, using image-conditioning pairs created from the renders of the NHPM dataset under various camera poses $\mathbf{c}$. The training objective for the geometric 2D prior follows the same training objective as a traditional SD model [69]:

$$L_{\text{simple}} = \mathbb{E}_{\mathbf{z}_0^n, \boldsymbol{\epsilon}, \boldsymbol{c}, t, \mathbf{y}_{\text{text}}, \mathbf{y}_{\text{id}}, \mathbf{y}_{\text{exp}}} \|\boldsymbol{\epsilon} - \boldsymbol{\epsilon}_{\phi_g}(\mathbf{z}_t^n, t, \boldsymbol{c}, \mathbf{y}_{\text{text}}, \mathbf{y}_{\text{id}}, \mathbf{y}_{\text{exp}})\|^2. \tag{5}$$

while the analogous training objective for the 2D texture prior can be derived by changing both superscript $n$ and subscript $g$ to $a$. Note that in Equation 5, $\mathbf{y}_{\text{exp}}$ indicates CLIP embeddings for the textual descriptor of the renders (i.e., camera view and subject's attributes) and does not convey face-identity. $\mathbf{z}_t^n$ indicates $\mathbf{z}_0^n$ noised at timestep $t$. Examples of generated prompt-to-images can be seen in Figure 2 as well as additional materials.

### 3.3 Geometry Generation

We represent an identity-specific geometry as a neural parametric head model composed of a deep marching tetrahedra (DMTET [77]) representation, additionally coupled with a set of identity-dependent facial expression latent codes. Our lightweight geometric representation produces highly detailed geometry and a broad range of expressions without notable identity drift, as shown by our experiments. Compared to explicit template-based approaches [92, 44, 43], it has the flexibility to dynamically modulate the local resolution of the mesh to capture high-frequency geometrical details. In other words, it adjusts the mesh resolution of specific regions of the face to adapt to the given subject and expression.

The DMTET geometry representation uses a deformable tetrahedral grid $\Gamma$ and a network $\Psi_g$ to generate a 3D asset [77]. We extend DMTET to learn multiple expressions at the same time. We model each facial expression with a learnable latent code $\mathbf{k}_{\text{exp}}^n \in \mathbb{R}^{d_{exp}}$ and design the network $\Psi_g(\Gamma, \mathbf{k}_{\text{exp}}^n)$ as a Transformer [81], parameterized by $\psi_g$ learnable parameters, that processes both the deformable grid and the expression information. During the training phase, we randomly select one of the potential expressions, estimate the signed distance function (SDF) for the underlying head, and use a differentiable marching tetrahedra layer to convert the implicit representation into the explicit surface mesh for that ID and expression, compelling the $\Psi_g$ model to learn a diverse set of expressions that are consistent with the identity at hand. We supervise optimization through an

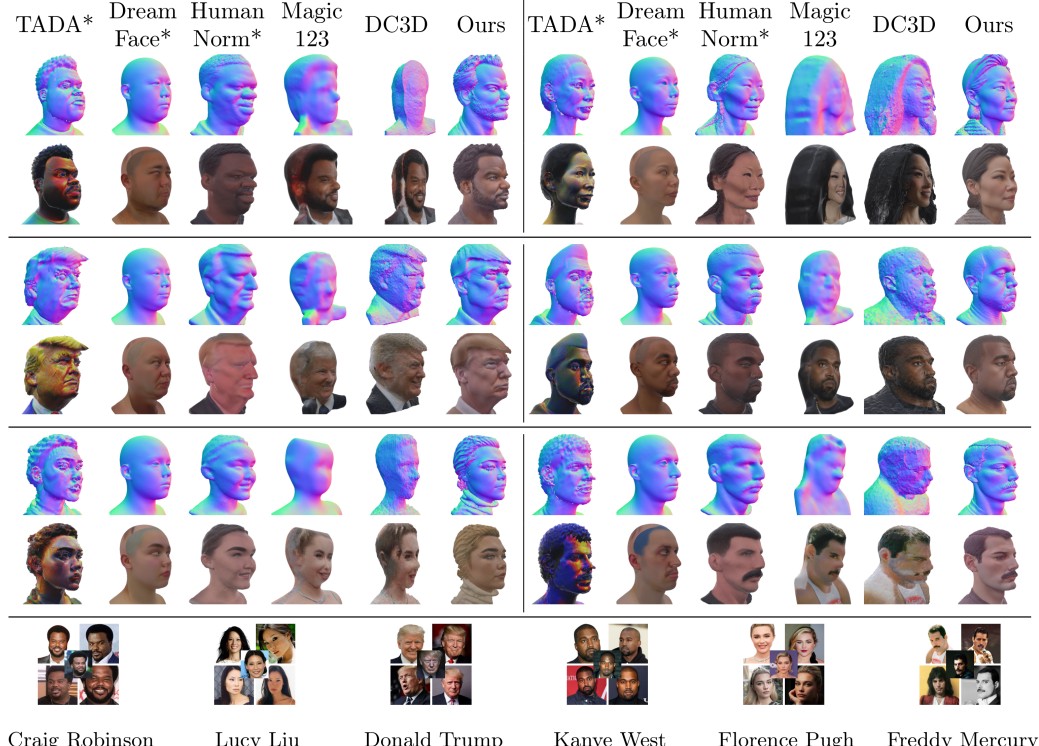

| TADA* | Dream Face* | Human Norm* | Magic 123 | DC3D | Ours | TADA* | Dream Face* | Human Norm* | Magic 123 | DC3D | Ours |

Craig Robinson          Lucy Liu          Donald Trump          Kanye West          Florence Pugh          Freddy Mercury

Figure 3: **Qualitative results for text-to-3D (*) and image-to-3D methods.** Methods are evaluated under the same text prompt and rendering conditions. DreamCraft3D is reported as DC3D. Geometry is displayed via normal maps in camera coordinates. Using only a small set of 5 images as conditioning, ID-to-3D achieves high geometric quality and realistic textures.

SDS loss [63], computed using the rasterized geometry model $\phi_g$ as 2D prior. We optimize the 3D representation $\theta_g$ as follows:

$$\nabla\mathcal{L}_{SDS}(\theta_g) = \mathbb{E}_{\mathbf{c},t,\epsilon}\left[\omega(t)(\epsilon_{\phi_g}(\mathbf{z}_t^n, \mathbf{y_{id}}, \mathbf{y_{exp}}, \mathbf{y_{text}}, t) - \epsilon)\frac{\partial g(\theta_g, \mathbf{c})}{\partial\theta_g}\right]; \theta_g = \left[\mathbf{k}_{exp}^n, \psi_g\right]. \quad (6)$$

In the above equation, $g(\theta_g, \mathbf{c})$ represents the normals of the rendered image, created using the differentiable render $g$ and the camera pose $\mathbf{c}$, and $\mathbf{k}_{exp}^n$ is the randomly sampled expression code. Lastly, we combine the SDS loss with a Laplacian regularizer, to encourage smooth surfaces.

### 3.4 Texture Generation

Given a trained geometry model $\theta_g$, we model the texture appearance $\theta_a$. To ensure ID-aligned expression generation, we follow an analogous parameterization for the expressions in the texture domain. We represent an identity-specific appearance as a neural parametric head model composed of a pseudo-albedo prediction network $\Psi_a$, coupled with a set of ID-dependent facial expression latent codes. We instantiate each expression as a learnable latent code $\mathbf{k}_{exp}^a \in R^{d_{exp}}$ and model $\Psi_a(\mathbf{k}_{exp}^a)$ as a Transformer trained to predict the spatially-varying reflectance in a UV-map representation [89], namely the albedo, roughness and specularity, for each ID and expression-specific texture. We use an off-the-shelf physically-based renderer [38]. Reflectance disentanglement is an ill-posed problem, and in the absence of prior data, we use camera and illumination randomization as a regularization constraint [18, 41]. On each iteration, we sample random environment illumination maps, augmented with random Y-axis rotations. During training, we randomize the sampling of the latent expression and deploy the albedo model $\phi_a$ SDS loss to optimize $\theta_a$:

$$\nabla\mathcal{L}_{SDS}(\theta_a) = \mathbb{E}_{\mathbf{c},t,\epsilon}\left[\omega(t)(\epsilon_{\phi_a}(\mathbf{z}_t^a, \mathbf{y_{id}}, \mathbf{y_{exp}}, \mathbf{y_{text}}, t) - \epsilon)\frac{\partial g(\theta_a, \mathbf{c}, \mathbf{l})}{\partial\theta_a}\right]; \theta_a = \left[\mathbf{k}_{exp}^a, \psi_a\right]. \quad (7)$$

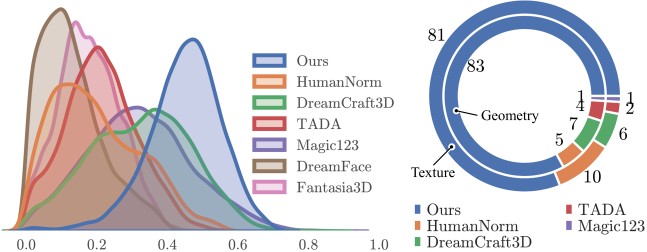

| | Texture ↓ | Geometry↓ |
|---|---|---|
| Fantasia3D* [10] | 252 | 159 |
| DreamFace* [92] | 188 | 145 |
| TADA* [44] | 215 | 118 |
| HumanNorm* [31] | 164 | 98 |
| Magic123 [64] | 162 | 180 |
| DreamCraft3D [79] | 205 | 165 |
| **Ours** | **153** | **85** |

Figure 4: **(Left) Identity Similarity Distribution** between "in-the-wild" images and renderings of 3D heads. **(Right) Comparative Preference Survey** on texture quality (outside) and geometry quality (in). We report % of preferences.

Table 1: **Frechet Inception Distance** measures the geometric and texture quality for the generated 3D heads. "*" indicates text-to-3D SDS pipelines.

where $g(\theta_a, \mathbf{c}, \mathbf{l})$ represents the pseudo-albedo maps created using a differentiable render $g$ and a sample camera pose and lighting condition.

## 4 Experiments

We assess the efficacy of ID-to-3D as a specialized method for ID-driven expressive human face generation in different scenarios and report comparative analysis against state-of-the-art text-to-3D and image-to-3D generation pipelines. Further analysis, implementation details and discussion can be found in additional material.

### 4.1 Identity Generation

We benchmark ID-to-3D against several state-of-the-art methods in the domain of SDS-based 3D face asset generation. Specifically, we consider Fantasia3D [10], three recent and similar methods specialized in text-to-human avatar generation (i.e., Human-Norm [31], TADA [44], DreamFace [92]) and two methods that leverage both text and images to create 3D assets (i.e., Magic123 [64], DreamCraft3D [79]). In order to compare together text-to-3D and image-to-3D methods, we select a test benchmark of 40 celebrity names, suggested by ChatGPT, covering actors, sports, and media personalities. We automatically download 25 images for each identity from BingImages [1]: 5 images to use as input and 20 to use as references for our comparisons. For all methods, we use the same textual prompt "a DSLR face portrait of..." and the same input images.

**Qualitative Comparisons.** Results of existing methods are reported in Figure 3 under the same lighting and rendering conditions. The 3D assets created by ID-to-3D show realistic texture, sharp fine-grained details, and ID fidelity, capturing facial characteristics of the input identity without relying heavily on often ambiguous text prompts or naively lifting 2D images in the 3D domain.

**Quantitative Comparisons: Identity Similarity Distribution.** We perform a quantitative evaluation on all the evaluated models using similarity ID. We assess the ID fidelity of the 3D assets using the CosFace similarity metric [83, 40]. We center and align the 3D objects and collect renders in a wide range of camera positions (i.e. elevation $[-15°, +15°]$ and rotation $[-65°, +65°]$). We measure the cosine similarity between the ID features of each render and a set of 20 in-the-wild images of the same identity, used as reference. We report the distribution of the identity similarity for each method in Figure 4 (Left). Note that the variance of the distribution correlates with the ID consistency of the 3D object across viewpoints. Despite being able to generate realistic skin textures, DreamFace cannot create hair or eyes by design, resulting in the lowest average similarity score. DreamCraft3D and Magic123 leverage the input image to achieve a photorealistic front-facing camera render, but struggle to create 3D consistent heads, resulting in distributions associated with the highest variance. As clearly visible from Figure 4, ID-to-3D is capable of creating realistic and consistent 3D heads, reporting the lowest variance and the highest similarity score.

**Quantitative Comparisons: FID and User-Study.** The evaluation of the generated 3D geometries and textures is performed using the Frechet Inception Distance (FID) metric [28]. For texture quality, the FID is calculated between the renderings and the images from Stable Diffusion V1.5 [69]. For geometry quality, FID is determined by comparing normal maps with those extracted from the NHPM test set [24]. To further substantiate our analysis, we conduct two user preference surveys, comparing our method with the 4 strongest baselines for the generation of texture and geometry. We compare

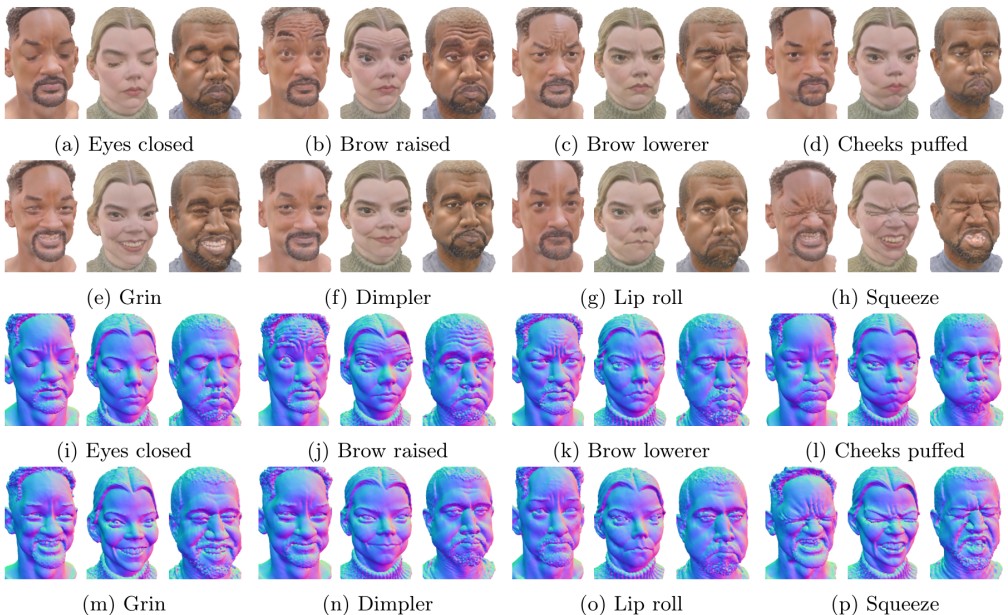

(a) Eyes closed     (b) Brow raised     (c) Brow lowerer     (d) Cheeks puffed

(e) Grin     (f) Dimpler     (g) Lip roll     (h) Squeeze

(i) Eyes closed     (j) Brow raised     (k) Brow lowerer     (l) Cheeks puffed

(m) Grin     (n) Dimpler     (o) Lip roll     (p) Squeeze

Figure 5: **ID-to-3D expression diversity**. Renderings and normal maps in camera coordinates are taken for 3 identities: *Will Smith*, *Anya Taylor Joy*, and *Kanye West*. Our method achieves fine-grained geometry carving and high-quality texture generation, realistically reproducing various skin tones.

the best 2 performing text-to-3D and image-to-3D methods against ID-to-3Din the user survey, as this enabled us to gather more responses. As shown in Table 1 and Figure 4 (Right), FID metrics and user evaluations report aligned results. Our model achieves the lowest FID scores and the highest user preference in both geometry and texture generation, showcasing together the overall stronger performance of ID-to-3Das a human-specific geometry and texture generator.

## 4.2 Expressive ID-conditioned Generation

ID-to-3D is specifically designed to create complex, uncommon, and subtle expressions with a level of details not previously achievable using existing SDS methods. In Figure 5 we showcase the unique ability of our method to generate a wide range of expressions that remain identifiable and yet identity-consistent.

**Quantitative Analysis: Identity Similarity Distribution Across Expressions.** We test the ID consistency across views and expressions by reporting the cosine similarity between the ID features of a reference render (i.e., neutral expression and front-facing camera) and the ID features captured across the remaining expressions and view points. Figure 6 (Right) shows the distribution of ID similarity computed for a set of 10 subjects. ID-to-3D consistently produces 3D assets with high ID similarity and low variance.

**Quantitative Analysis: Expression Variety Visualization.** A core characteristic of ID-to-3D is its robust ability to produce a wide range of unique and vivid expressions. In this section, we provide a visualization of this expression diversity. We select a subset of 13 expressions and 10 subjects, extracting for each 3D head a set of renders in a range of 9 camera poses. Then we extract the ID features for each render and project them into 2D points using t-SNE. As visible in Figure 6 (Middle), plotting these points clearly shows the heterogeneity of the generated expressions and identities. Firstly, renders associated with the same identity are grouped in distinct clusters. Furthermore, within each cluster, we observe how there is a local variation of ID features as they adapt their response to different expressions.

## 4.3 ID-conditioned Text-based Customization

The unique characteristics of our model allow for the customization of 3D objects without identity or expression drift. Figure 7 and 2 (Right) showcase how a variety of alterations can be imposed on the

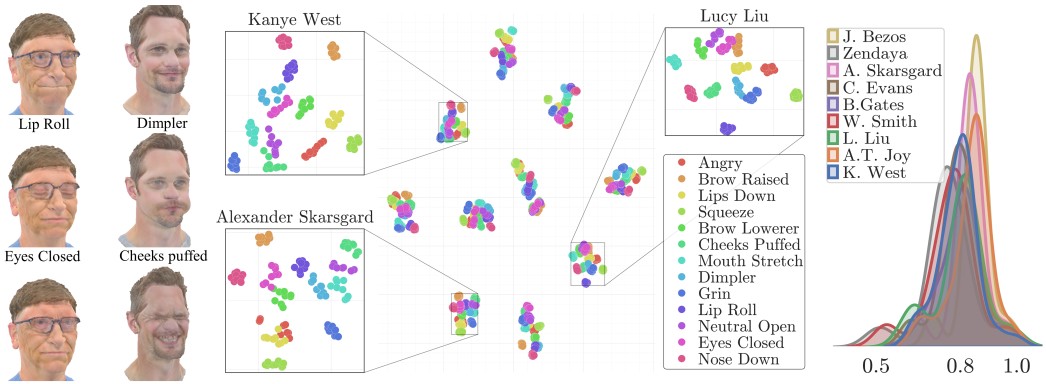

Figure 6: **Expressions analysis.** ID-to-3D creates a variety of expressions with robust ID consistency. **(Left) Visualizations** of different expressions for 2 identities (i.e. *Bill Gates*, *Alexander Skarsgard*). **(Middle) Expression diversity**. t-SNE plot visualizing the ID embeddings computed considering different camera poses, expressions, and subjects. Different identities and expressions are clustered separately. **(Right) Identity similarity distributions** between neutral-pose and remaining expressions. Rendered with [2].

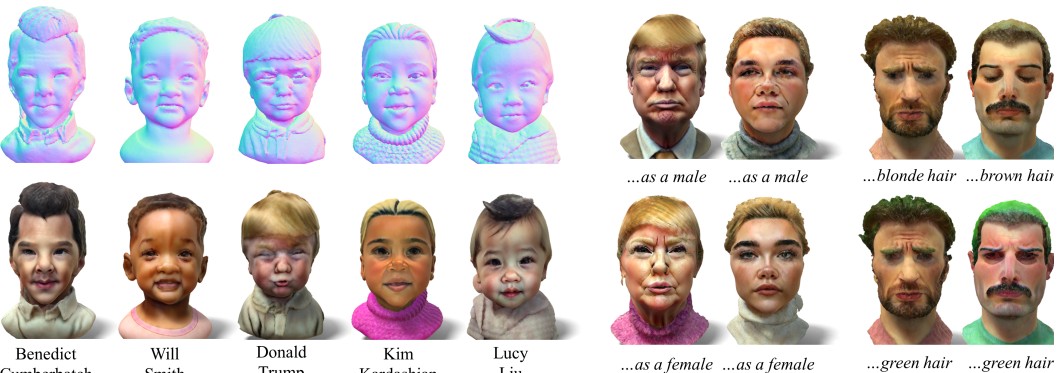

Figure 7: **Identity-Consistent Editing. (Left) De-aged 3D heads** generated using different identity conditioning and the textual prompt: *"...as a cute baby"*. Normal maps are displayed in world coordinates next to photorealistic renderings. **(Right) Geometry and texture editing** with text prompts. ID-to-3D edits appearance and geometric features in an ID-consistent manner.

object's geometry, given the appropriate text conditioning while maintaining the subject's general physiognomy. In particular, ID-to-3D displays aligned geometry and appearance even after editing, preserving the ability to convey different expressions (e.g. from "eyes closed" to "brow lowerer", from "neutral" to "squeeze") even after facial hair textures have been altered via textual prompt.

Figure 8 provides evidence on the text-guided editing capability of our model for richer and more complex textual prompts. In particular, we showcase text prompts associated with various hairstyles, head accessories, and face shape changes driven by different ethnic backgrounds. Note that our method can interpret and exploit text-based inputs not addressed in previous works (e.g. id-driven changes in "aging", "gender", "heritage"). Even when using the exact same text prompt (e.g. "A woman with African-American heritage, ... 70's hairstyle.") our model generates unique identity-consistent assets that simultaneously align with the text and retain the characteristic facial features of the input ID. Our approach is able to address practical scenarios and opens new avenues for expressive text-guided editing of 3D assets.

## 5 Conclusion and Ethical Considerations

**Limitations.** Despite setting a new state-of-the-art, we acknowledge ID-to-3D limitations: **(1)** The generalization capacity is constrained by the employed face embedding network [17], the pretrained

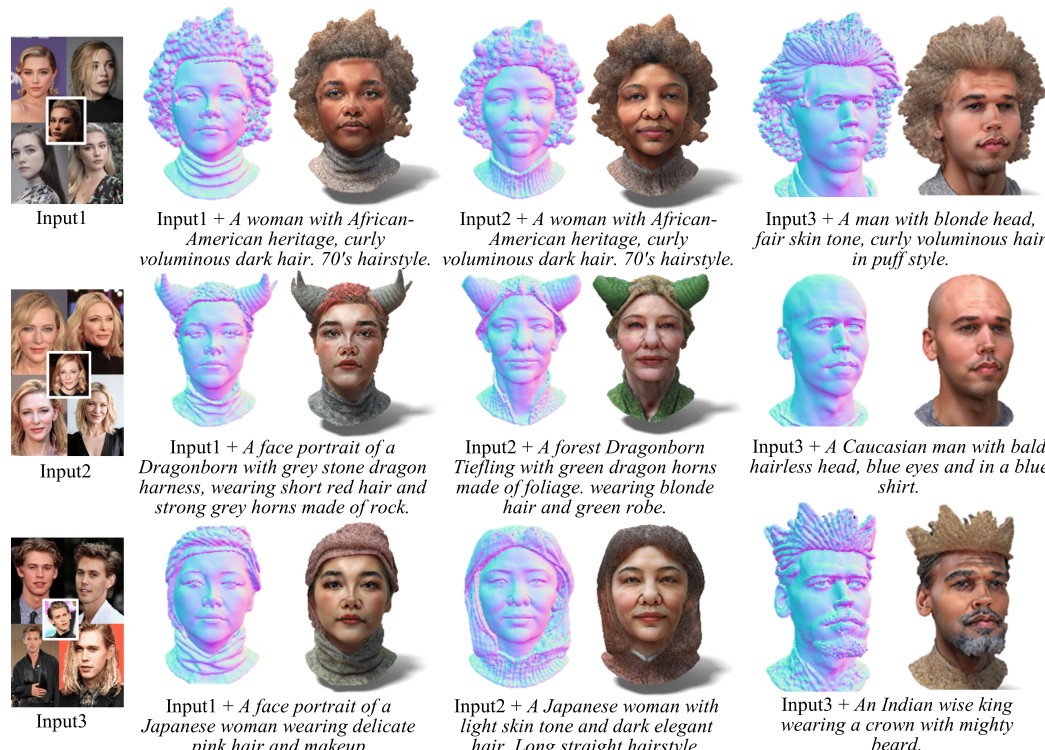

Figure 8: **Identity-Consistent Editing with Rich Textual Prompts.** ID-to-3D generates ID-consistent assets that reflect both subtle and significant changes in geometry and appearance described in a detailed textual input. Normal maps are displayed in world coordinates next to photorealistic renderings and input prompts.

large-scale diffusion models [69, 88], and the finetuning of the 2D guidance models on a dataset of relatively small size, which might result in inaccurate representation biases and biases due to dataset imbalances; **(2)** the texture-guidance model results might come short of photorealism. The generated textures mimic the albedos from the [24] dataset, which contains expression-rich data but only low-resolution UV diffuse albedo maps; **(3)** the lack of specific optimization for physically bounded textures and geometries might occasionally produce unnatural exaggerated facial characteristics, hence we call them pseudo-albedo, despite producing render-ready assets; **(4)** the computational resources needed for PBR rendering hinder potential applications in video-driven problems.

**Societal Impact.**   Technological advancements in automatic 3D human generation have various beneficial applications, but also raise important ethical considerations about representation and potential misuses. We advocate for responsible research and take the following steps to mitigate unauthentic reconstructions: **(1)** we restrict training data to the facial region only. **(2)** we advocate for the replacement of text-based prompts with larger ID embeddings that are sometimes lacking [90] but entail significantly less bias than methods trained only on celebrity datasets and text [65].

**Conclusion.**   In this work, we present ID-to-3D , a novel method for expressive 3D head asset generation from one or more face images. Our method deploys a novel human parametric expression model in tandem with specialized geometry and albedo guidance, not only to create intricately detailed head avatars with realistic textures but also to achieve strikingly ID-consistent results across a wide range of expressions, setting a new benchmark in comparison to existing SDS techniques. Without having to rely on 3D captured datasets that are expensive to collect and typically biased, and without being constrained on a specific geometry template, our method can be employed by a broad range of subjects, with different features such as skin tone and hairstyle.

**Acknowledgements.**   S. Zafeiriou and part of the research was funded by the EPSRC Fellowship DEFORM (EP/S010203/1), EPSRC Project GNOMON (EP/X011364/1) and Turing AI Fellowship (EP/Z534699/1).

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
