# OpenReview forum: "ID-to-3D: Expressive ID-guided 3D Heads via Score Distillation Sampling"
_NeurIPS.cc/2024/Conference — NeurIPS 2024 poster_

### Official Review · Reviewer_fD9E · 2024-06-16

**Soundness:** 4
**Presentation:** 3
**Contribution:** 3
**Rating:** 7
**Confidence:** 5

**Summary:**

The paper introduces a novel method for generating 3D human heads guided by both identity and textual descriptions. The proposed method employs face image embeddings and textual descriptions to optimize a neural representation for each subject. By leveraging task-specific 2D diffusion models as priors and a neural parametric representation for expressions, the method achieves high-quality results while requiring minimal training. Extensive experiments demonstrate the proposed method could generate higher-quality 3D facial assets than state-of-the-art methods.

**Strengths:**

1. The proposed method is novel and promising in the field of 3D facial asset generation. The use of task-specific 2D diffusion models as priors for 3D head generation is a novel technique that reduces the dependency on large 3D datasets. The method incorporates a neural parametric representation to disentangle expressions from the identity, which is a reasonable way to manage the complexity of facial dynamics and ensure identity consistency in generated 3D heads.

2. Experiments evaluation is strong and demonstrates the superiority of the proposed method over existing methods. The quantitative analysis and visualization are impressive. The visual results also clearly show the advantage over state-of-the-art methods. Detailed metrics and visualization highlight the method's ability to produce high-quality, identity-consistent 3D models, showcasing significant improvements in terms of details and textures. The evaluation covers various aspects such as generalization to new identities, handling of different ethnicities, and avoidance of oversmoothing, providing a thorough validation of the method's effectiveness.

3. The generated 3D models are of high quality, with detailed textures and geometry. The method ensures that the identity of the input image is well-preserved in the 3D output. The versatility of the method is also demonstrated, which provides practical utility beyond simple model generation.

**Weaknesses:**

1. The generated 3D head models, while detailed, sometimes appear exaggerated and more like caricatures than realistic human faces. This diminishes the method's applicability in scenarios where photorealism is critical. Addressing this issue would involve refining the model to balance between preserving identity and achieving realistic facial features.

2. The performance and quality of the method on significantly larger and more complex datasets are not extensively discussed. Evaluating and demonstrating the scalability of the approach would enhance its credibility and applicability. Discussing potential limitations when scaling up and providing strategies to handle larger datasets would be valuable additions. It would be interesting to discuss or show what could be further improved if larger datasets are incorporated.

3. The writing can be improved for better clarity and accessibility. For instance, it should be explicitly described that each identity requires a training stage. This would help readers understand the necessity and implications of the training process. Additionally, simplifying and clarifying the description of the multi-stage pipeline would make the methodology more accessible to other researchers and practitioners.

**Questions:**

See the weaknesses section.

**Limitations:**

Limitations are discussed in the paper.

---

> ### Author Rebuttal · Authors · 2024-08-06
>
> # 5 Response to reviewer `fD9E`
> We thank the reviewer `fD9E` for recognizing our work as "novel and promising", acknowledging its "superiority [...] over existing methods" and appreciating its "versatility". Below we address the proposed concerns about the limitations of our method and its potential extensions.  **We regularly refer to the general response above and the one-page pdf provided with the figures**
>
> ## 5.1 Bias towards exaggerated features.
> Indeed, the current pipeline exhibits a bias towards exaggerated features. We identify 3 main causes of this phenomenon:
>
> 1. **Dataset size**. We train our 2D guidance model on the NPHM dataset, which is characterized by a relatively small number of subjects. Consequently, it may under-represent the wide spectrum of human head shapes and textures, resulting in output that may appear cartoonish. We provide a breakdown of dataset diversity in Fig.E and a more detailed comment in Sec.1.3 of the general response.
>
> 2. **Dataset texture quality**. While the NPHM dataset excels in expressive details, it lacks high-quality texture scans. As a result, the generated 3D model textures may appear unnatural in some cases. Examples of NPHM renders and our generated 2D textures can be seen in Fig.G and H respectively, and a more detailed discussion can be found in Sec.1.1 of the above general response.
>
> 3. **Geometry generation pipeline**. Our method employs a sequence of random camera poses to guide the geometry generation process. However, this can introduce point-of-view biases, leading to certain portions of the 3D model being overemphasized.
>
> Considering the above observations, we theorize that using a larger-scale dataset could address the issue by improving the photorealism of the generated 3D head models, and recognize that it represents one of the immediate next steps to further advance our work.
>
> ## 5.2 Discussion on scaling up to larger and more complex datasets.
> Scaling up our SDS pipeline to exploit larger or more complex datasets is an interesting problem, as long as such datasets are available.\
> We identify 2 potential next steps:
> - **Extending the current NPHM dataset** by acquiring more scans to overcome its distribution bias as discussed in the reply above. This extension would not entail any adjustments to the method but would improve its generalization capabilities.
> - **Leverage more complex data types** by combining datasets with temporal signals (e.g. video) and very high-quality textures (e.g. lightstage data). Merging different data types into our pipeline would require method adjustments. One solution could be to employ a proxy representation, such as 3DMMs or a UV texture template, to allow for easy segmentation of semantics (i.e. hair/face subdivisions, etc.), that could be easier to link with large-scale SD textual embeddings. In this direction, we identify the recently released AVA256 ( 13th June 2024) [1] as a good candidate dataset to use in future work.
>
> ## 5.3 Improve the writing  for better clarity and accessibility.
> We appreciate and agree with the reviewer's recommendations to improve the manuscript. We will revise and update the text as suggested to improve clarity.
>
> [1] [https://about.meta.com/realitylabs/codecavatars/ava256](https://about.meta.com/realitylabs/codecavatars/ava256)

---

> > ### Comment · Reviewer_fD9E · 2024-08-13
> > **Maintain rating**
> >
> > The rebuttal addressed some of my concerns. I recommend the authors explain the exaggerated caricature-like style in the paper and maybe need to change the "realistic" claims. I would maintain my initial rating.

---

> > > ### Author Response · Authors · 2024-08-13
> > >
> > > We thank the reviewer for the feedback. We appreciate the suggestions and will ensure to comment on the caricature-like style in the limitations section of the paper and revise the claims as suggested.

---

### Official Review · Reviewer_G3ux · 2024-07-10

**Soundness:** 3
**Presentation:** 3
**Contribution:** 3
**Rating:** 6
**Confidence:** 4

**Summary:**

The paper presents a novel technique for creating 3D human head models from a single real-world image, guided by identity and text. This method is based on compositionality and uses task-specific 2D diffusion models as optimization priors. The authors extend a base model and fine-tune only a small training parameters to create 2D priors for geometry and texture generation. Additionally, the method utilizes a neural parametric representation for expressions, allowing the creation of highly detailed geometry and albedo textures.

**Strengths:**

1. The method demonstrates significant innovation in the field of 3D head generation, particularly in generating high-quality models without the need for large-scale 3D datasets.
2. The capability to generate 3D models with disentangled expressions is a notable advancement, as this has been a challenge in previous research.

**Weaknesses:**

1. The paper should discuess dataset diversity. Does the training dataset used in this study possess enough diversity to prevent potential biases?
2. There's a lack of analysis on the robustness of ID embeddings. Are these identity embeddings robust enough to accurately depict identity features across various expressions and poses?
3. Some citations are missing, specifically at line 191 where template-based approaches require references.
4. Is it logical for geometry diffusion to be fine-tuned using style-transfer methods? It could be reasonable if a pre-trained standard SD model like RichDreamer[1] is utilized.
5. Some results aren't convincing; for instance, in the last row of Fig9 within supplementary materials, reconstruction seems to lose eye-catching hair details.
6. Supplementary materials have been placed in a separate zip file instead of being attached to the main paper, which might breach some submission rules.

[1] RichDreamer: A Generalizable Normal-Depth Diffusion Model for Detail Richness in Text-to-3D

**Questions:**

* Discussing the diversity of the dataset during training.
* Discussing the robustness of id embeddings
* Correcting citations.
* Explaining why fine-tune using a geometry diffusion model with style-transfer instead of another geometry diffusion model.

**Limitations:**

The authors acknowledged the limitations and deliberated on the possible negative impacts of the suggested technology on society.

---

> ### Author Rebuttal · Authors · 2024-08-06
>
> # 4 Response to reviewer `G3ux`
> We appreciate the reviewer's thorough feedback and the definition of our work as a "significant innovation in the field of 3D head generation". We provide all the requested clarification and experiments below. **We regularly refer to the general response above and the one-page pdf provided with the figures.**
>
> ## 4.1 Discussion on dataset diversity.
> Please refer to Sec.1.3 of the above general response.
>
> ## 4.2 Analysis of the robustness of ID embeddings.
> We provide analyses on the robustness of the ID embeddings w.r.t. expressions and camera poses in Fig.C and Fig.D, which we will include in our work.\
> We use our training dataset (NPHM dataset). For each identity and expression, we collect renders for 9 different camera rotation angles [-60°, -45°, -30°, -15°, 0°, 15°, 30°, 45°, 60°] and extract their identity embedding (ArcFace [1]). Examples of renders are visible in Fig.G. We use a neutral pose with a 0° rotation angle as a reference.\
> Note that, per definition, a similarity score above 0.5 signifies the same identity [1].
> - **The impact of expressions on ID embeddings (Fig.C)** is isolated by computing the cosine similarity between the neutral reference and all the remaining expressions captured with a rotation angle of 0°.
> - **The impact of the camera pose on ID embeddings (Fig.D)** is isolated by computing the cosine similarity between the neutral reference and the neutral expression captured with all the 9 possible rotation angles.
>
>
> As visible from Fig.C and D, ArcFace reliably captures identity features across various expressions and poses, showcasing robust behavior even for extreme expressions (e.g. "Squeeze" Avg-SimID: 0.62) and substantial camera rotation (e.g. -60° Avg-SimID: 0.7).
>
>
> ## 4.3 Missing references.
> We added references to line 191 [2,3,4]
>
> ## 4.4 Initialization of the 2D models.
> ID-to-3D uses two ID-driven and task-specific guidance models during geometry and texture generation.\
> We create both geometry-oriented and texture-oriented models starting from the same pre-trained weights. This design choice avoids the introduction of two different initialization biases and has the goal of favoring ID consistency between the two finetuned representations (Fig.7 Additional, Fig.H rows 1 and 2).
>
> ## 4.5 Artifacts.
> Despite avoiding any Janus artifacts or major misalignments, failure cases of our ID-to-3D might involve slight misalignment between texture and geometry (e.g. Fig.9 last row in additional material).  We suggest that this is due to the lack of specific optimization for physically bounded textures and geometries and plan to improve on this portion of the pipeline in future work.
>
> [1] Arcface: Additive angular margin loss for deep face recognition. CVPR 2019\
> [2] DreamFace. SIGGRAPH 2023\
> [3] TADA. 3DV 2024\
> [4] FLAME: Learning a model of facial shape and expression from 4D scans. SIGGRAPH 2017

---

> > ### Comment · Reviewer_G3ux · 2024-08-14
> >
> > Thanks for the author’s response. I would raise the score to 6.

---

### Official Review · Reviewer_vm8B · 2024-07-11

**Soundness:** 3
**Presentation:** 3
**Contribution:** 3
**Rating:** 5
**Confidence:** 3

**Summary:**

This work proposes a new approach for the generation of 3D human heads, which enables guidance with identity, facial expressions, and text descriptions. The approach is structured around two principal components: 1) the authors fine-tune a previously established text-to-image diffusion model through LORA on a specialized dataset of 3D human head models, to obtain the 2D guidance with separated texture and geometric details. 2) the method executes the generation of geometry and texture in separate stages, utilizing the SDS loss to optimize the process. It considers specific designs, such as the learnable latent code of facial expression, to enrich the geometry and texture details. It obtains better performance than several existing works.

**Strengths:**

1. The proposed method is reasonable. The finetuning of the diffusion model on a specific 3D head dataset provides better guidance for geometry and texture.
2. It supports various types of conditional inputs, including identity, expression, and text description, thereby enriching its versatility in application.
3. It obtains better performance than several existing works.

**Weaknesses:**

1. The generated head is of low visual quality. We are aware that text-to-image diffusion models are capable of producing very high-resolution images. However, the learned facial textures (resolution, clarity) showcased in this study are relatively poor. In comparison, other single-image 3D reconstruction methods, such as 3D GAN inversion combined with technologies like NeRF, can achieve very high visual quality. The text guided 3D portrait generation method, Portrait3D (siggraph 24), is also of high visual quality.
2. The innovativeness of the method is modest. Fine-tuning diffusion models on specific datasets and using SDS as a supervisory loss for 3D modeling are both fairly common practices. The methodological innovation in this study seems insufficient.
3. The expressions and text-guided editing scenarios demonstrated in the experimental section are quite basic (such as "eyes closed," "brow lowerer," "de-aged"), which limits their practicality. It is suggested to showcase more practical editing effects, such as changes in hairstyle or face shape, richer text-based guidance, to better understand its editing performance.

**Questions:**

There are a few questions that should be addressed in the rebuttal, please see paper weakness for more information.

******** after rebuttal
Thanks for providing additional experiments, that addressed some of the concerns. I raised my rating to borderline accept, mainly for its contribution of providing a relatively complete method for simultaneously controlling facial ID, expressions, and characteristics.

Yet, the generated head, especially the texture, is still low in quality. Although this may be influenced by the dataset used, it is still a limitation of the method as there is currently no better dataset available (per the author's rebuttal), and it is also unlikely to construct a higher-quality dataset. Is there any other possible ways to further improve the image quality?

**Limitations:**

yes

---

> ### Author Rebuttal · Authors · 2024-08-06
>
> # 3 Response to reviewer `vm8B`
> We thank the reviewer for the feedback. We thoroughly clarify and respond to all the concerns raised.  **We regularly refer to the general response above and the one-page pdf provided with the figures.**
> ## 3.1 Discussion and comparisons on the quality of the generated textures.
> We address the 3 comparisons asked by the reviewer in the following:
>
> ### 3.1.1 Comparison with Stable-Diffusion text-to-image models.
> In Fig.H, we provide comparisons of our specialized 2D models against two well-established Stable-Diffusion text-to-image models usually used in SDS pipelines [7,8].
> As visible, these Stable-Diffusion models have three main shortcomings that limit their applicability to the task at hand (i.e. creation of ID-driven 3D heads with expression control):
> - __Low ID retention.__ The models struggle to consistently create outputs of specific identities since they rely only on textual prompts.
> - __Low expressivity.__ The use of natural language to enforce expression conditioning is ineffective. The models overlook expression-related prompts to boost photorealism.
> - __Inconsistent Lighting.__ The models generate a wide range of lighting conditions, to enhance photorealism and artistic effect. This complicates the separation of lighting and albedo contributions when creating renderable-ready assets with SDS.
>
> Please refer to Sec.1.1 of the above general response for a more in-depth discussion on the impact of the NPHM dataset bias on texture quality.
>
> ### 3.1.2 Comparison with 3D-aware image synthesis methods.
> Please refer to Sec.1.2 of the above general response for a description of the advantages of our 3D representation against single-image 3D reconstruction methods that use 3D GAN inversion combined with NeRF.
>
> ### 3.1.3 Comparison with Portrait3D (siggraph 24).
> We acknowledge this concurrent work (as per NeurIPS guidelines) published on 19 July 2024 in the ACM Transactions on Graphics (TOG), Volume 43, Issue 4. We will include and discuss it in the main manuscript.
>
> ## 3.2 Limited Innovation.
> As noted by `G3ux`, `HnyJ` and `fD9E`, our approach introduces significant advancements in 3D head generation, addressing key challenges in the field. In particular, we present to the reviewer 2 overlooked contributions, further substantiating the innovativeness of our model:
>
> - __Production of editable identity-driven 3D heads with expression control via SDS.__ To the best of the author's and other reviewers' knowledge, we propose the first SDS model producing identity-driven, editable, and highly detailed 3D heads with expression control from in-the-wild images of subjects without the need for large-scale 3D datasets, which allow us to set a new SoTA for human heads generation via SDS.
>
> - __Design of a neural parametric expression model compatible with SDS pipelines.__ This methodological innovation allows ID-to-3D to disentangle expressions from identity in 3D assets, a challenge in SDS research [1,2]. The control over facial expressions ease editing while ensuring consistent identity preservation across diverse 3D models.
>
> ## 3.3 Expression and text-guided editing capability.
> We address the 2 concerns raises below:
> ### 3.3.1 Expression Conditioning.
> > The expressions [...] demonstrated in the experimental section are quite basic.
>
> Our expression controls can convey extreme expressions (e.g. ’squeeze’ and ’cheeks puffed’), as well as handle subtle changes from the neutral reference (e.g. ’dimpler’ and ’lip roll’) with unprecedented levels of geometric detail and expression-conditioned wrinkles (Fig.5 and 6 of the main paper).\
> As highlighted by `G3ux` and `fD9E`, our method goes beyond basic scenarios and shows promising results compared to relevant literature:
> 1) The capability to generate 3D models with disentangled expressions has been a challenge in previous SDS research [1, 2]
>
> 2) Managing the complexity of facial dynamics while ensuring identity consistency in generated 3D heads is a well-known problem in literature, with specialized methods like [3, 4, 5, 6] all struggling to create expression-driven wrinkles.
>
> ### 3.3.2 Text-Guided Editing.
> > The [...] text-guided editing scenarios demonstrated in the experimental section are quite basic.
>
> As requested, we provide evidence on text-guided editing capability of our model beyond basic scenarios. Fig.A presents the text-guided editing scenarios suggested by the reviewer. In particular, we showcase:
> - 11 unique rich text prompts associated with 6 hairstyles (Fig.A1-8, A10, A11).
> - 2 head accessories (Fig.A3, A4, A9).
> -  5 face shape changes driven by 3 different ethnicities (Fig.A1, A2, A5, A6, A9).
>
> Note that :
> -  Our method can interpret and exploit text-based inputs not addressed in previous works (e.g. id-driven changes in 'aging', 'gender', 'heritage').
>
> -  Even when using the exact same text prompt (Fig A1, A2) our model generates unique identity-consistent assets that simultaneously align with the text and retain the characteristic facial features of the input ID.
>
> Our approach is able to address practical scenarios and opens new avenues for expressive text-guided editing of 3D assets.
>
> [1] HumanNorm. CVPR 2024.\
> [2] HeadSculpt. NeurIPS 2023.\
> [3] Learning neural parametric head models. CVPR 2023.\
> [4] DECA: Detailed Expression Capture and Animation. SIGGRAPH 2021.\
> [5] SMIRK: 3D Facial Expressions through Analysis-by-Neural-Synthesis. CVPR 2024.\
> [6] GANHead: Towards Generative Animatable Neural Head Avatars. CVPR 2023.\
> [7] Stable Diffusion 2.1. [https://huggingface.co/stabilityai/stable-diffusion-2-1](https://huggingface.co/stabilityai/stable-diffusion-2-1).\
> [8] DreamLike-Photoreal 2.0. [https://huggingface.co/dreamlike-art/dreamlike-photoreal-2.0](https://huggingface.co/dreamlike-art/dreamlike-photoreal-2.0).

---

> ### Comment · Reviewer_vm8B · 2024-08-13
> **Will the code and models be open sourced?**
>
> Will the code and models be open sourced?

---

> ### Author Response · Authors · 2024-08-13
> **Code publicly available**
>
> Yes, we confirm that the code and the models will be made publicly available.

---

### Official Review · Reviewer_HnyJ · 2024-07-12

**Soundness:** 3
**Presentation:** 3
**Contribution:** 3
**Rating:** 5
**Confidence:** 4

**Summary:**

**Summary
This paper focuses on the task of 3D head generation. Specifically, the authors first extend a traditional diffusion model to a text-to-normal version and a text-to-albedo version with ID-aware and expression-aware cross-attention layers. Then, with the trained diffusion models, the authors optimize a neural parametric head model with a score distillation sampling loss. Extensive experiments demonstrate that the proposed method outperforms existing text-to-3D and image-to-3D methods in terms of 3D head generation. However, I have some concerns about this paper. My detailed comments are as follows.

**Strengths:**

**Positive points
1.	The authors introduce the first method for arcface-conditioned generation of 3D heads with score distillation sampling loss.
2.	The proposed method can also achieve ID-conditioned text-based 3D head editing (e.g., age editing, changing hair color and gender).

**Weaknesses:**

1.	Although the proposed method generates a similar geometry to the input identity, the synthesized texture appears much unrealistic. What might be the cause of this phenomenon? Additionally, some implicit 3D representations, like NeRF [A-C] and 3DGS [D-F], can model high-fidelity surfaces for 3D heads. Why do the authors choose DMTET over these representations? It would be better if the authors could provide more discussion about the above questions.
2.	The authors use five images as identity references for each 3D head. How is this optimal number determined? What is the relationship between identity similarity and the number of reference images? Quantitative results in terms of this should be provided.
3.	There are some misalignments between Table 1, Figure 3, and Figure 4. For example, Fantasia3D is included only in Table 1 but does not appear in Figure 3 or the right column of Figure 4..
4.	In the original paper of Fantasia3D [G], the proposed method can only synthesize 3D assets given a text prompt as input. How do the authors adjust this methods to generate 3D heads when conditioned on specific identity?
5.	For each 3D head asset, the authors extract identity features from multiple RGB images. How are these features combined? Are they added or concatenated? It would be helpful to provide more details about this operation.
**Minor issues
1.	On page 7, line 263, there is a missing space between “ID-to-3D” and “as”.
**Reference
[A] NeRF: Representing Scenes as Neural Radiance Fields for View Synthesis. ECCV 2020.
[B] Implicit and Disentangled Face Lighting Representation Leveraging Generative Prior in Neural Radiance Fields. TOG 2023.
[C] Geometry-enhanced Novel View Synthesis from Single-View Images. CVPR 2024
[D] 3D Gaussian Splatting for Real-Time Radiance Field Rendering. SIGGRAPH 2023.
[E] Photorealistic Head Avatars with Rigged 3D Gaussians. CVPR 2024.
[F] Relightable Gaussian Codec Avatars. CVPR 2024.
[G] Disentangling Geometry and Appearance for High-quality Text-to-3D Content Creation. ICCV 2023.

**Questions:**

Please refer to the weakness section.

**Limitations:**

Yes.

---

> ### Author Rebuttal · Authors · 2024-08-06
>
> # 2 Response to reviewer `HnyJ`
> We thank the reviewer for recognizing the novelty of our method and appreciating the experimental section. Below we respond to the doubts put forward by the reviewer. **We regularly refer to the general response above and the one-page pdf provided with the figures.**
> ## 2.1 Details about the method.
> We provide the additional requested clarifications:
> ### 2.1.1 Cause of unrealistic texture.
> Please refer to Sec.1.1 of the above general response.
> ### 2.1.2 Choice of 3D representation.
> Please refer to Sec.1.2 of the above general response.
>
> ## 2.2 Relationship between identity similarity and number of reference images.
> We analyze the relationship between identity similarity and the number of reference images in Fig.B. We consider 40 identities, 25 in-the-wild images for each subject, and extract for each image its identity embedding (ArcFace) [1]. For each identity, we consider the center of the distribution as representative of its facial features. We report the similarity between the center of the distribution and the mean ArcFace created with a subset of N number of reference images.
> The plot shows the averaged trend for 40 identities (blue line) together with its standard deviation (light blue).
> The trend reaches a plateau after 20 images, while 5 images is enough to reach an identity similarity of more than 0.95 for all the IDs considered. In our experiments, we selected 20 images to use as references for our comparisons and kept 5 images to use as input to our method, ensuring a good trade-off between identity similarity retention and practicality.
>
> ## 2.3 Misalignments in Figures 3,4 and Table 1.
> We clarify the misalignments:
> - We perform a quantitative evaluation on all the evaluated models using similarity ID (Fig.4 left) FID on geometry and texture (Table 1).
> - We compare the best 2 performing text-to-3D and image-to-3D methods against ID-to-3D in the user survey (Fig.4 right), as this enabled us to gather more responses.
> - We omit the worst-performing method (Fantasia3D) in the qualitative comparisons (Fig.3) due to space constraints.
> We included a qualitative comparison against Fantasia3D in Fig.F. We will amend the revised version of the paper to include comparisons and clarification.
>
> ## 2.4 Creation of Fantasia3D assets.
> Stable Diffusion textual embeddings struggle to represent a specific identity but have knowledge of named celebrities included in the training data, which can be generated by prompting "name + surname". To compare ID-to-3D and text-to-3D methods, we create a dataset of celebrity names suggested by ChatGPT and use the textual prompt "A DSLR portrait of [name + surname]" to create Fantasia3D / TADA / Human-Norm / DreamFace 3D assets.
>
> ## 2.5 Identity features concatenation.
> For each image, we extract the ArcFace identity embedding after cropping and centering. The identity embeddings are then concatenated and processed by a shallow 2-layer MLP to match the dimension of the text features in the pretrained diffusion model. This representation serves as identity conditioning for the geometry-oriented and albedo-oriented 2D diffusion models.
>
> [1] Arcface: Additive angular margin loss for deep face recognition. CVPR 2019

---

> > ### Comment · Reviewer_HnyJ · 2024-08-13
> >
> > The authors have addressed my concerns. I have decided to maintain my score as a Borderline accept.

---

### Author Rebuttal · Authors · 2024-08-06

We sincerely appreciate the reviewers’ insightful feedback, which acknowledges the novelty of our method (G3ux, HnyJ, fD9E) and the quality of our experimental results (G3ux, HnyJ, fD9E, vm8B). The reviewers recommended additional experiments and visualizations to emphasize strengths, address limitations, and highlight future improvements. We have diligently conducted all the suggested experiments, as detailed in our responses to both general and reviewer-specific comments. **To prevent confusion with the original submission’s figures, we have labeled the new figures in the attached document using alphabetical letters.**

# 1 General Response
## 1.1 Generation of unrealistic textures. [ Reviewers `HnyJ`, `vm8B`, `fD9E` ]
ID-to-3D creates the textures of the expressive 3D heads with 2D guidance specialized in pseudo-albedo generation. This 2D model reliably generates ID-driven images under consistent and diffuse lighting conditions (Fig.H row 1), while also allowing direct control over the subject's expression -- two features absent in previous text-to-image SD models [1, 2]  (Fig.H rows 3 and 4).
We train our model using the NPHM dataset [3], which is rich in expression data and geometric quality. \
Nevertheless, we acknowledge 2 of its main shortcomings that affect the synthesized texture quality:
- **Dataset bias.** Due to its limited size, it is inherently prone to biases, especially when compared to standard datasets used for SD training (e.g. Lion 5B [4], Lion-Face 20M [5]).
- **Dataset texture details.** The dataset provides scans with only UV diffuse albedo maps, which exhibit a relatively low level of texture photorealism (as visible from its renders in Fig.G).

Despite these drawbacks, NPHM remains the best choice of its type in the public domain, and given higher-quality data, our method could greatly benefit.
*The generated 3D textures resemble the NPHM suboptimal 2D albedos encountered during the training of the texture-guidance model. Nevertheless, this limitation is substantially outweighed by the capability to create images and 3D assets with control over the subject's expression and identity.*

## 1.2 Alternative 3D representations. [ Reviewers `HnyJ`, `vm8B` ]
In our 3D representation, we use DMTET for geometry generation and a Transformer predicting spatially varying reflectance during texture generation. This 3D representation offers 2 key advantages:

- **Combines benefits from implicit and explicit 3D representations** [6]. This allows the generation of high-frequency details and expression-dependent wrinkles while directly producing render-ready 3D heads (i.e. textured 3D meshes).
- **Disentangles geometry and appearance generation.** It allows a separate optimization of geometry and appearance, ensuring flexibility in editing and mitigating the generation of artifacts.

Compared to other methods:
- **3D-aware image synthesis methods** [7,8,9,10] focus on generating novel 2D views of input images by using 3D GAN inversion combined with NeRF-based representation. They generally struggle to handle side or back views effectively and do not produce textured meshes that can be used as-is in downstream applications. We consider this line of work orthogonal to ours.

- **NeRF-based methods** [11] jointly optimize density and RGB color for a given scene. Similarly, **3DGS-based methods** [12, 13, 14] optimize at once explicit 3D gaussians and their spherical harmonics for appearance. Although they achieve high visual quality when trained on pixel-perfect views of a scene, when used in SDS pipelines they both a) struggle to perform effective surface recovery of high-frequency details and b) train intertwined geometry and appearance representation that cannot be easily manipulated.

We agree that other representations also offer interesting possibilities for future research.
In particular, 3DGS shows very good results in human 3D reconstruction from lightstage data [14] or high-quality videos [13]. How to extend the 3DGS representation to achieve photorealistic generation from in-the-wild images using SDS is an open but interesting line of research.

## 1.3 Dataset bias. [ Reviewers `G3ux`, `fD9E` ]
We appreciate the insightful questions and agree with the importance of discussing dataset diversity and minimizing potential biases.
We provide a detailed breakdown of the NPHM dataset [3] in Fig.E. During the 2D guidance training, we use gender, age, ethnicity, and hairstyle as textual prompts to guide generation. As a result, our method generates diverse identities, ethnicities, and ages (Fig.A, Figures of the main paper). We are aware and acknowledge the limitations inherent in a small-sized dataset, such as the underrepresentation of minorities, and we revise the main manuscript to add clarifications and highlight this critical issue.

[1] Stable Diffusion 2.1. https://huggingface.co/stabilityai/stable-diffusion-2-1.

[2] DreamLike-Photoreal 2.0. https://huggingface.co/dreamlike-art/dreamlike-photoreal-2.0.

[3] Learning neural parametric head models. CVPR 2023.

[4] LAION-5B: An open large-scale dataset for training next generation image-text models. NeurIPS 2022.

[5] General Facial Representation Learning in a Visual-Linguistic Manner. CVPR 2022.

[6] Deep Marching Tetrahedra. NeurIPS 2021.

[7] Implicit and Disentangled Face Lighting Representation Leveraging Generative Prior in Neural Radiance Fields. TOG 2023.

[8] Geometry-enhanced novel view synthesis from single view images. CVPR 2024.

[9] Efficient Geometry-aware 3D Generative Adversarial Networks. CVPR 2022.

[10] PanoHead: Geometry-Aware 3D Full-Head Synthesis in 360°. CVPR 2023.

[11] NeRF: Representing Scenes as Neural Radiance Fields for View Synthesis. ECCV 2020.

[12] 3D Gaussian Splatting for Real-Time Radiance Field Rendering. SIGGRAPH 2023.

[13] Photorealistic Head Avatars with Rigged 3D Gaussians. CVPR 2024.

[14] Relightable Gaussian Codec Avatars. CVPR 2024.

---

### Comment · Area_Chair_eNJE · 2024-08-13
**Response to Authors**

Dear Reviewer G3ux,

The authors have responded to the questions that you raised in your review. The discussion period with the authors is soon coming to a close, at 11:59 PM AOE Aug 13, 2024. There is still time to respond to the authors. It would be great if you could acknowledge that you have read the authors' response, engage in discussion with them and update your final score.

Best,
AC

---

### Decision · Program_Chairs · 2024-09-25

**Decision:**

Accept (poster)

**Comment:**

This paper presents a method for ID and expression preserving 3D head synthesis using pre-trained diffusion models. To accomplish this task, the authors propose to finetune a 2D diffusion model into a normal and an albedo prediction models using a smaller dataset using NPHM with ID and expression controls added in. The authors propose to them use these diffusion models to learn a DMTET style-mesh and texture representation for heads, which can be relit and controlled for expressions. The results show improvements in terms of geometric quality and expression control versus the existing SOTA approaches using pre-trained diffusion models and SDS loss.

Reviewers noted the low texture quality and cartoonish appearance of the generated results as a limitation, which the authors acknowledge as stemming from the diffuse albedo of the models in NPHM. Other reviewers also noted the lack of comparisons to recent Nerf and Gaussian splatting based approaches, which are more photorealistic in quality, which the authors correctly pointed out as not being relightable. Notwithstanding these limitations, the approach does present the best in class accuracy, and is a step forward in advancing the generation of ID and expression controlled 3D faces from diffusion models. Hence, all reviewers recommend accepting this paper. The AC concurs and recommends acceptance.